

# Tracking neonicotinoids following their use as cotton seed treatments

Kristina L. Kohl[1], Lauren K. Harrell[2], Joseph F. Mudge[1],
Seenivasan Subbiah[1], John Kasumba[1], Etem Osma[3], Apurba K. Barman[4]
and Todd A. Anderson[1]

[1] Department of Environmental Toxicology, Texas Tech University, Lubbock, TX, USA
[2] Texas A&M AgriLife Extension Service, Wharton, TX, USA
[3] Biology Department, Erzincan Binali Yildirim University, Erzincan, Erzincan Province, Turkey
[4] Department of Entomology, University of Georgia, Tifton, GA, USA

Corresponding author
Todd A. Anderson,
todd.anderson@ttu.edu

## ABSTRACT

Neonicotinoids are a leading class of insecticides on the global market, accounting for nearly 25%. They are widely used in both agricultural and residential settings. Causing neuron failure by irreversibly binding to the insect nicotinic acetylcholine receptor, neonicotinoids offer broad spectrum efficacy against a variety of pests. However, because they are non-selective with regard to insect species, there has been some concern with neonicotinoid use over threats to pollinators such as honeybees, and potential indirect effects to migratory waterfowl as a result of invertebrate prey population depletion. In order to study occurrence and fate of neonicotinoids (thiamethoxam and imidacloprid), we analyzed cotton leaves on plants grown from neonicotinoid-treated seeds and corresponding soil samples between cotton rows. Neonicotinoid concentration data from cotton leaves appears to be consistent with the claim that seed treatments protect plants for 3–4 weeks; by 30 days post-planting, neonicotinoid concentrations fell, in general, to 200 ng/g or lower. This represents about a 10-fold decrease from plant concentrations at approximately 2 weeks post-planting. It was found that neonicotinoids used as seed treatments remained present in the soil for months post planting and could be available for runoff. To that end, 21 playa wetlands were sampled; 10 had at least one quantifiable neonicotinoid present, three of which were classified as grassland or rangeland playas, two were urban, and the remaining five were cropland playas. In several instances, neonicotinoid concentrations in playas exceeded EPA chronic benchmarks for aquatic invertebrates.

## INTRODUCTION

Neonicotinoid pesticides, a class of seven chloro-nicotinic insecticides, represent one of the fastest growing segments of the insecticide market (*Simon-Delso et al., 2015*) with a broad spectrum of agricultural and residential applications (*Jeschke et al., 2011*). Application methods vary according to crop, but many are used as seed treatments (seed coatings). Neonicotinoids are moderately persistent in the environment; half-lives in soil

typically range from a few weeks to several months (*Sharma & Singh, 2014*), but can be much longer according to field studies (*Morrissey et al., 2015*). In aquatic systems, some neonicotinoids are susceptible to photolysis with half-lives as short as 1–2 days (*Bonmatin et al., 2015*); however, natural waters are usually turbid so these half-lives may be much longer.

Possible non-target consequences to pollinator species, particularly honeybees, as a result of neonicotinoid application have been a concern and focus of research to date (*Krupke et al., 2012*; *Tsvetkov et al., 2017*; *Samson-Robert et al., 2017*). A possible link between unintentional pollinator death and neonicotinoid use has resulted in restrictions on neonicotinoid applications in Europe (*Maxim & Van Der Sluijs, 2013*). Initially, use of thiamethoxam, clothianidin, and imidacloprid was restricted on crops "attractive to bees" across the European Union, but in 2018 that restriction was expanded to include all outdoor uses; thiamethoxam, clothianidin, and imidacloprid can only be applied in greenhouses.

Effects on pollinators are not the only concerns related to neonicotinoid use. Recently, potential dangers to aquatic invertebrates in wetlands have become an additional area of research focus (*Finnegan et al., 2017*). Subsequent indirect threats to more charismatic aquatic and terrestrial organisms as a result of impacts to aquatic invertebrates are of research interest (*Sánchez-Bayo, Goka & Hayasaka, 2016*). A possible connection between declines in insectivorous birds and reduced aquatic invertebrate populations has been observed (*Hallmann et al., 2014*); however, the potential impact on other species, such as migratory waterfowl, is less known.

Use of neonicotinoid seed treatments is gaining popularity due to increased selectivity for certain pests, long-term protection, and reduced use of active ingredient. It has been suggested that the use of neonicotinoids as a seed treatment, rather than as a drip treatment or foliar application may reduce some of the threat to pollinators, although this assertion is heavily debated (*Tomizawa & Casida, 2005*; *Girolami et al., 2009*; *Huff Hartz, Edwards & Lydy, 2017*); the negative impact of neonicotinoids on insect parasitoids and natural enemies has limited their use in IPM programs.

When applied to the seed, a portion of the active neonicotinoid ingredient is translocated, via the vascular tissue (xylem), throughout the whole plant, offering protection against piercing-sucking insects (*Girolami et al., 2009*; *Goulson, 2013*). Movement of neonicotinoids from the treated seed to the surrounding soil also occurs and is a function of irrigation and water movement within the soil (*Sánchez-Bayo et al., 2007*). As the biomass of the plant increases over the growing season, the concentration of neonicotinoids in the plant decreases (*Balfour et al., 2016*). Seed treatments are typically purported to protect the plant from target pests for 3–4 weeks after emergence (*Maienfisch et al., 2001*).

Cotton is an important crop in Texas, and specifically in the Southern high plains (SHP) region. Neonicotinoid seed treatments are well suited for cotton, because they offer broad spectrum protection against many sucking insects, such as thrips and aphids, which are common early season cotton pests (*Elbert et al., 2008*; *Maienfisch et al., 2001*). As such, neonicotinoids (particularly imidacloprid and thiamethoxam), applied as seed

treatments and via other methods, are widely used across the SHP. In a region like the SHP, where agriculture can be a risky undertaking due to weather conditions and a shorter production window, offering extra protection from pests at the beginning of a plants life can be crucial.

In the semi-arid SHP, playa wetlands are the main surface hydrological features. Existing mainly in their own individual watersheds, these dipressional wetlands act as catchment basins for most of the surrounding runoff, including runoff due to rain events and irrigation (*Belden et al., 2012*); the runoff may contain agrochemicals such as fertilizers and pesticides. Playa wetlands provide important habitat for a host of wildlife, including migratory birds and the aquatic invertebrates they depend on as a food source. Each year, millions of waterfowl, shorebirds, and landbirds winter in or pass through the SHP as part of the Central Flyway. Invertebrate density is a key factor impacting the carrying capacity of a playa for migratory bird populations (*Anderson & Smith, 1999*).

The initial goal of this study was to determine neonicotinoid (imidacloprid and thiamethoxam) concentrations in soil and cotton leaves as a result of seed treatment use. However, the study also provided opportunities to estimate and assess neonicotinoid runoff from cotton fields when used as seed treatments to playa wetlands.

## MATERIALS AND METHODS

Cotton leaf and field soil samples were obtained through cooperation with Texas A&M AgriLife Extension Service. Cotton was cultivated in-field at five locations in West Texas; viz. Halfway, Kress, Lamesa, Wall, and Chillicothe. Thiamethoxam and imidacloprid treated seeds were used, with all seed coatings containing approximately 0.25 mg of active ingredient per seed. Planting occurred in May of 2014 (the exact date varied by location). Four consecutive replicate blocks were grown at each location; each block contained a set of thiamethoxam-treated seeds, imidacloprid-treated seeds, and one or two sets of control seeds grown in rows within the block. The number of rows per treatment within a block varied between three and four, depending on the field. Spaces were left between each block to signify the beginning of a new replicate. Fields were irrigated via center pivot sprinkler (a common practice in the area) and no additional neonicotinoids were applied to the plants throughout the growing season.

### Sampling

Leaf samples were collected from plants during the month of June 2014. Chillicothe, Lamesa, and Wall sites were sampled twice; Kress and Halfway were only sampled once. When two samplings occurred, dates of collection were roughly 1–2 weeks apart. Whole leaves were taken from the plant, bagged and transported to the lab, where they were frozen until analysis. At the first sampling, samples were composites of several small leaves from the same plant in order to have sufficient tissue mass. Subsequent samples consisted of a lone, mature (5th) leaf.

Soil cores were collected in June and at the end of September/beginning of October, 2014. Three sites (Chillicothe, Lamesa, and Wall) were sampled twice, while two sites (Kress and Halfway) were only sampled once. When two samplings occurred, dates of

collection were roughly 1–2 weeks apart. An 11 cm core of soil was collected at a distance of approximately 15 cm from a plant perpendicular to the cotton row (the typical distance between planted rows was 75 cm); the top three cm of the core was discarded. By collecting 15 cm from the planted row, the goal was to avoid collecting any un-germinated seeds or remaining seed coatings in the sample. Soils were placed in sample bags, and returned to our laboratory, where they were stored at −20 °C until analysis.

Playa water samples were collected in 2014 from 21 playas in the northern region of the Texas SHP. Playas were in watersheds classified as cropland (10 sites), grassland (seven sites), or urban (four sites). Playas were classified as cropland if they received most of their water from drainage and runoff from agricultural lands currently in use for crop production. Playas that were in rural locations but did not receive runoff from farmed land were classified as grassland. These playas may have been located on rangeland or simply on non-utilized grasslands. Urban playas were located in parks and golf courses within the city limits of Lubbock, TX. Approximately 250 mL of water was collected from each playa, in water 15–30 cm deep. The amber sample jars were capped, stored in coolers, and transported back to Texas Tech University, where they were stored frozen until analysis.

## Sample analysis

Cotton leaf samples were allowed to dry for 72 h between weighings, then roughly cut into small pieces and placed in 15-mL conical tubes with 12 mL of acetonitrile. To determine dry weight, soil samples were weighed and placed in the hood for 48 h to dry, then weighed again. The samples were then transferred to 50-mL conical tubes and 25 mL of acetonitrile added to each sample. All samples were agitated on a shaker table for 2 h. After shaking, samples were left for 24 h, at which point they were centrifuged at 2,000 rpm for 10 min. As much acetonitrile as possible was decanted from each sample into a 15-mL conical tube and volumes were recorded. After evaporating to dryness, samples were reconstituted in one mL of 1:1 high performance liquid chromatography (HPLC)-grade acetonitrile and HPLC-grade water. Samples were filtered using syringeless auto-sample vials (0.45 μm glass microfiber filter) and analyzed as described below. Acetamiprid was used as a surrogate spike for quality assurance purposes, as it was not used in the experimental fields and therefore not expected to occur in the leaf and soil samples. Every fifth cotton leaf and soil sample was spiked using a stock solution of acetamiprid in methanol. Spike-recovery tests for neonicotinoids in blank plant and soil samples indicated that extraction efficiency was quantitative and reproducible.

Neonicotinoids were extracted from water samples using solid phase extraction (SPE). A 50-mL water sample was first filtered using a 0.45 μm Nylon syringe filter. Burdick and Jackson $C_{18}$ SPE cartridges (1,000 mg) were conditioned with five mL of methanol, followed by five mL of Milli-Q® water. The filtered water samples were passed through the conditioned cartridges and dried under vacuum for 5 min. Cartridges were then eluted with five mL of acetonitrile in one-mL increments. Samples were evaporated to dryness

and reconstituted in 0.5 mL of 1:1 HPLC-grade acetonitrile and HPLC-grade water, then transferred to two-mL glass vials for HPLC analysis.

Given that the cotton test plots represented a controlled scenario where they were expected to contain known analytes, we used conventional HPLC with UV detection rather than a more definitive technique like liquid chromatography-mass spectrometry (LC-MS) for the soil and plant samples. Analyte concentrations were determined using an Agilent 1100 series HPLC with a UV detector, and a Thermo Scientific BDS Hypersil $C_{18}$ column (250 × 4.6 mm; three μm particle size) for analyte separation. A mobile phase of 70% water and 30% acetonitrile was used at a flow rate of 0.5 mL/min. All solvents were HPLC grade. Sample injection volume was 25 μL and sample run time was 25 min. A wavelength of 270 nm was used for the detection and quantification of imidacloprid; 254 nm was used for thiamethoxam. In hindsight, we should have also tested for clothianidin, a metabolite of thiamethoxam produced in cotton plants (*Nauen et al., 2003*).

Initially, playa water samples were analyzed with an Agilent 1100 series HPLC-UV using conditions identical to those described above for leaf and soil samples with the exception that clothianidin and acetamiprid were added to the analyte list. Confirmatory analyses were conducted using LC-MS. For those analyses, HPLC parameters were as follows: column—Gemini NX-$C_{18}$, three μ particle size, 150 mm length × two mm diameter; mobile phase A = water + 0.1% formic acid, mobile phase B = acetonitrile + 0.1% formic acid; mobile phase composition (A:B, v/v) was 100:0 at 0 min, 30:70 at 3 min, 15:85 at 6 min, 10:90 at 9 min, 50:50 at 12 min, and 100:0 from 14 to 15 min; flow rate = 100 μL/min; injection volume = 20 μL. For MS parameters, we used heated electrospray ionization in the positive mode with selected reaction monitoring. Parent and product ions for thiamethoxam, imidacloprid, acetamiprid, and clothianidin were 292 → 211 and 132, 256 → 209 and 175, 223 → 126 and 56, and 250 → 132 and 169, respectively.

## Statistical analysis

Because we sampled plots in which specific neonicotinoids had been used, residue data from those field plots (soil and plant samples) were normally distributed. We used ANOVA to test for site (location) effects followed by a Tukey's post hoc test. We used simple *t*-tests to determine effects of time (soil concentration differences between 1st and 2nd sampling point) within a site. Thiamethoxam residue data for playa samples were normally distributed, however, data for the other neonicotinoids were not normal and followed a typical pattern (logarithmic) for environmental residues where non-detects are frequent. We used ANOVA (α = 0.05) to compare thiamethoxam concentrations and/or total neonicotinoids by playa type.

## RESULTS AND DISCUSSION

Generally, plant and soil sample concentrations were quite variable, even among replicates in the same treatment field for a given sampling date. Detection of treatment compounds was isolated to the respective treatment replicate samples in all instances; thiamethoxam was never detected in samples from imidacloprid-treated plots and vice

**Table 1 Mean[1] cotton leaf tissue concentrations of thiamethoxam and imidacloprid (±standard error of the mean) as determined by HPLC[2] in 2014.**

| Site (planting date) | Sampling date (days post planting) | Thiamethoxam[3] (ng/g) | Imidacloprid[3] (ng/g) |
|---|---|---|---|
| Chillicothe (5/19) | 6/4 (15) | 2,704 ± 836 | 2,313 ± 406 |
|  | 6/19 (28) | 72 ± 39 | 180 ± 91 |
| Lamesa (5/17) | 6/10 (23) | 6,840 ± 2,912 | 1,783 ± 600 |
|  | 6/18 (30) | 218 ± 147 | 247 ± 136 |
| Wall (5/23) | 6/18 (25) | 3,234 ± 317 | 1,436 ± 320 |
|  | 6/23 (30) | 18 ± 13 | 5.0 ± 0 |
| Halfway (5/6) | 6/11 (35) | 18 ± 13 | 8.0 ± 3.3 |
| Kress (5/13) | 6/3 (20) | 10,618 ± 3,365 | 15,100 ± 4,885 |

**Notes:**
[1] We used the leaf quantitation limit (five ng/g) for non-detect samples in calculating the mean.
[2] The MDL for thiamethoxam and imidacloprid in leaf tissue extracts was two ng/g.
[3] Based on the locations sampled twice and assuming losses followed a linear relationship, the estimated thiamethoxam and imidacloprid dissipation half-lives in cotton leaves were 4.3 and 4.5 days, respectively.

versa. Neither treatment compound was detected in any soil or plant sample from control plots.

## Cotton leaves

Overall, average cotton leaf concentrations of imidacloprid by site and sampling date varied from non-detect in all four samples (Wall, 6/23/2014) to 15,100 ng/g (ppb) (Kress, 6/3/2014) (Table 1). Thiamethoxam averages ranged from 18 ppb (Wall, 6/23/2014; Halfway, 6/11/2014) at 30-35 days after planting to >10,600 ppb (Kress, 6/3/2014) at 20 days after planting.

For the sites where cotton leaves were sampled twice, samples were collected no more than 15 days apart, with the final sample collected between one and 1.5 months after planting. In general, concentrations of individual neonicotinoids were similar across all sites, with respect to date of sampling. That is, leaf concentrations at initial sampling were higher than those at subsequent dates. Average thiamethoxam concentrations tended to be slightly higher than average imidacloprid concentrations when compared by site and date of sampling, suggesting that the more water soluble thiamethoxam is more readily translocated through the plant in xylem. However, we observed a large amount of sample variation (as reflected by the relatively high standard errors among sample replicates), so any data interpretation should be made with that in mind.

Imidacloprid concentrations significantly decreased between the first and second samplings ($p < 0.05$ for all sites). Average concentrations generally fell by an order of magnitude to approximately 250 ppb or less. In the second round of samples, non-detects were frequent; imidacloprid was not detected in any of the second round of samples from Wall. Decreases in thiamethoxam concentrations were significant at two of the three sites: Wall and Chillicothe, but not at Lamesa. While Wall and Chillicothe had similar concentrations to each other at both the first and second samplings, average

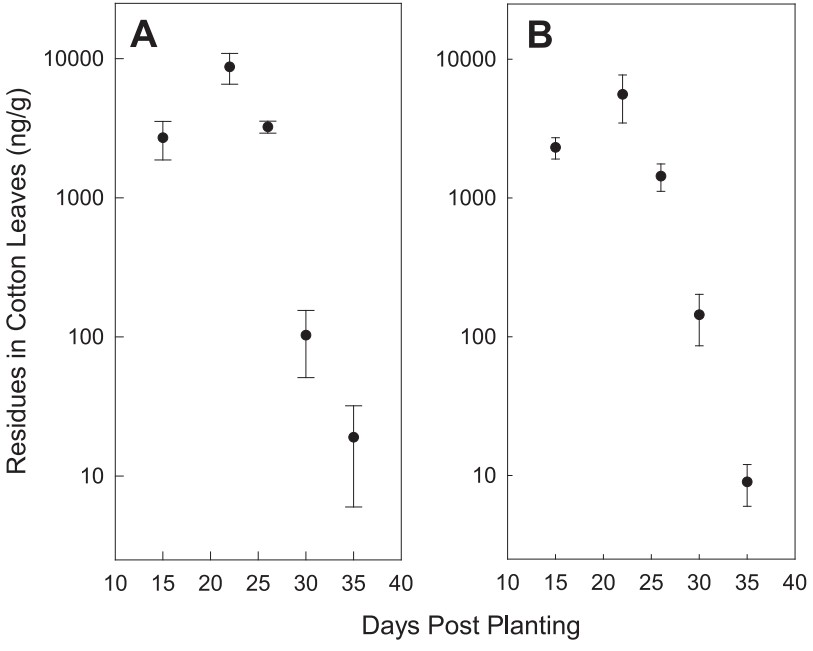

**Figure 1** Mean (for all locations) leaf tissue concentrations of thiamethoxam (A) and imidacloprid (B) compared with number of days since planting. Error bars are the standard error of the mean.

concentrations at Lamesa were higher at both time points, due to a relatively large variation among the replicates.

Although Halfway was only sampled once, 35 days after planting, the average concentrations of imidacloprid and thiamethoxam seem to correspond to the values of samples from other sites collected at a similar time since planting (second samplings of Chillicothe, Lamesa, and Wall). Kress was also only sampled once, 20 days after planting. Average concentrations of both imidacloprid and thiamethoxam at Kress were higher than samples from other locations taken at comparable dates (first sampling of Chillicothe, Lamesa, and Wall). The average imidacloprid concentration in cotton leaves was 7–11 times higher at Kress than at the other three sites and the differences were statistically significant; the average thiamethoxam concentration was about two to four times higher at Kress than at Chillicothe, Lamesa, or Wall, although the differences were not statistically significant.

When neonicotinoids are applied via seed treatment, and no other applications of neonicotinoids occur, there is a finite amount of active ingredient available to be taken up by a plant. When plants first emerge and biomass is low, the concentration of neonicotinoids is relatively high. As the plant increases in biomass, it effectively dilutes that concentration, resulting in a lower concentration of active ingredient in the plant tissue (*Balfour et al., 2016*). Based on the locations sampled twice and assuming losses followed a simple linear relationship, the thiamethoxam and imidacloprid dissipation half-lives in cotton leaves were 4.3 and 4.5 days, respectively.

Our neonicotinoid concentration data from the cotton leaves appears to be consistent with the claim that seed treatments protect plants for 3–4 weeks (*Greenberg, Liu &*

**Table 2 Mean soil concentrations of thiamethoxam and imidacloprid (±standard error of the mean) as determined by HPLC[1] in 2014.**

| Site (planting date) | Sampling date (days post planting) | Thiamethoxam (ng/g) | Imidacloprid (ng/g) |
|---|---|---|---|
| Chillicothe (5/19) | 6/4 (15) | 20 ± 8.8 | 35 ± 13 |
| | 6/19 (28) | 33 ± 17 | 28 ± 12 |
| Lamesa (5/17) | 6/10 (23) | 65 ± 29 | 162 ± 144 |
| | 9/29 (134) | 4.9 ± 0.5 | 5.2 ± 4.0 |
| Wall (5/23) | 6/18 (25) | 14 ± 5.2 | 16 ± 2.1 |
| | 9/29 (128) | 9.4 ± 1.2 | 8.2 ± 2.5 |
| Halfway (5/6) | 10/1 (147) | 20 ± 4.5 | 98 ± 47 |
| Kress (5/13) | 10/1 (140) | 12 ± 1.2 | 38 ± 34 |

Notes:
[1] The MDLs for thiamethoxam and imidacloprid in soil extracts were 1.0 and 0.5 ng/g, respectively.

*Adamczyk, 2009*); by 30 days post-planting, concentrations fell, in general, to 200 ppb or lower (Fig. 1). This represents about a 10-fold decrease from concentrations at approximately 2 weeks post-planting. If concentrations of 5–10 ppb in plant tissue are sufficient to protect against pests, as previously reported (*Greenberg, Liu & Adamczyk, 2009*), then the duration of protection may extend out to 5–6 weeks.

## Soil

All soil samples from treated plots had quantifiable amounts of their respective seed treatment compound (either imidacloprid or thiamethoxam), regardless of treatment, field, or time of sample collection (Table 2). No treatment soil samples had detectable amounts of neonicotinoids other than those they were treated with. Neither thiamethoxam nor imidacloprid were detected in any soil samples corresponding to control (untreated seeds) plots.

Comparisons were conducted between average soil imidacloprid and thiamethoxam concentrations across sampling dates at individual sites. Imidacloprid concentrations in Wall were the only ones that decreased, or changed, significantly between sampling dates ($p < 0.05$). Imidacloprid concentrations did not significantly vary with regards to time in Lamesa or Chillicothe. It is important to note that the average soil concentration in Lamesa on 6/10/2014 was much higher than all other imidacloprid samples from comparable sampling dates, due to one extreme (outlier) concentration of 593 ppb. Because this concentration is so much higher than any other detected imidacloprid concentration, it seems plausible that a seed casing was collected in the soil core, resulting in an artificially inflated concentration. Even with this high average, there was no statistically significant difference between imidacloprid concentrations in June and September sampling dates.

There was no statistically significant difference in thiamethoxam concentrations with regards to time at Wall, Lamesa, or Chillicothe. The general lack of change in soil neonicotinoid concentrations was expected given the relative high half-lives of imidacloprid and thiamethoxam in soil. It is interesting to note, however, that it was a set of imidacloprid samples that decreased significantly, given that the half-life of imidacloprid in ideal soil conditions is up to 228 days, whereas thiamethoxam's half-life is

up to 72 days (*Morrissey et al., 2015*). September samples were collected approximately 4.5 months post planting, well past the 72 day maximum half-life of thiamethoxam. This lack of expected degradation suggests that the release of neonicotinoids from the seed coating does not happen quickly, but rather slowly as the seed germinates and then as the seed casing breaks down. Since 80% of the coating remains in the soil (*Goulson, 2013*), seed treatments could result in accumulation long-term (*Jones, Harrington & Turnbull, 2014*). Finally, neonicotinoid half-lives determined under controlled laboratory conditions may be different than observed degradation in environmental conditions which also include dissipation "losses" from leaching and runoff.

While neonicotinoid seed-treatments may offer long term protection, it is also important to consider that the soil residue results from our study and others (*Jones, Harrington & Turnbull, 2014*) show that active ingredient is moving from the seed coating to the soil, and remaining there for months after planting. Less than 5% of imidacloprid used in cotton seed treatment is taken up by the germinating seed, leaving a significant mass available to move elsewhere (*Sur & Stork, 2003*). The presence of neonicotinoids in soil allows for the movement of those chemicals to other environmental compartments such as surface water (*Hladik, Kolpin & Kuivila, 2014*), through runoff and leaching (*Wettstein et al., 2016*).

Tier I type assessments of agrochemical runoff are used to estimate worst-case scenarios and assess the potential for ecological threats based on available data. Simple models are frequently used to help accomplish the task of determining whether pesticide concentrations may potentially exceed environmental benchmarks. One screening mechanism, often termed the "Back-of-the-Envelope" calculation, is based largely on the solubility of the chemical. This screening forms the basis of the GENeric estimated exposure concentration model (*Parker, Jones & Nelson, 1995*) in use today. We used this "Back-of-the-Envelope" calculation to assess neonicotinoid runoff potential when used as seed coatings, supplementing the traditional watershed/pond characteristics for those more relevant to SHP cotton agriculture, watersheds, and playa wetlands.

Estimates of playa area, depth, watershed (runoff) size, cotton seed drop rate, and active ingredient per seed, as well as compound solubility were needed for the model. Values of 6.3 ha for playa area and 55.5 ha of runoff were used, based on the average playa size in the SHP and average playa watershed size (*Haukos & Smith, 1994*). A playa depth of 0.25 m was used based on average cropland and grassland playa depths from a previous SHP study (*Luo et al., 1997*). Cotton seed planting rate was assumed to be 65,000 seeds per acre (0.4047 ha). Neonicotinoid extraction from treated seeds used to grow cotton revealed an average of 0.25 mg of active ingredient (neonicotinoid) per seed.

Based on these assumptions, it was estimated that 2.23 kg of active ingredient would be present in a 55.5 ha watershed, due to cotton seed treatment. Because all of the analyzed neonicotinoids have a water solubility greater than 100 ppm (for example, thiamethoxam = 4.1 g/L and imidacloprid = 0.61 g/L), 5% of the applied chemical (111 g in this case) in the watershed is assumed to be in the runoff (*Parker, Jones & Nelson, 1995*). In this situation, with a 6.3 ha playa (0.25 m deep), the resulting maximum neonicotinoid concentration due to seed treatments would be 7.1 ng/mL (ppb).

**Table 3 Neonicotinoids in playa water samples collected in 2014.**

| Playa type | | Thiamethoxam | Imidacloprid | Acetamiprid | Clothianidin |
|---|---|---|---|---|---|
| Urban | 4 sites | 50% | 0% | 0% | 0% |
| | (Low) | 0.3 ng/mL | ND | ND | ND |
| | (High) | 3.8 ng/mL | ND | ND | ND |
| Mean (total neonicotinoids) ± standard error = 1.1 ng/mL ± 0.9 | | | | | |
| Crop | 10 sites | 40% | 20% | 0% | 20% |
| | (Low) | 0.6 ng/mL | 1.1 ng/mL | ND | 0.1 ng/mL |
| | (High) | 3.1 ng/mL | 1.8 ng/mL | ND | 1.1 ng/mL |
| Mean (total neonicotinoids) ± standard error = 1.0 ng/mL ± 0.4 | | | | | |
| Range | 7 sites | 29% | 0% | 14% | 29% |
| | (Low) | 0.5 ng/mL | ND | 3.5 ng/mL | 0.2 ng/mL |
| | (High) | 0.6 ng/mL | ND | 3.5 ng/mL | 0.2 ng/mL |
| Mean (total neonicotinoids) ± standard error = 0.7 ng/mL ± 0.6 | | | | | |

**Notes:**
Frequency of detection, as well as low and high concentrations for individual neonicotinoids are indicated. ND, not detected. The limit of detection for neonicotinoids in water was approximately 0.05 ng/mL. The limit of quantitation was 0.1 ng/mL.

## Playa samples

Of the 21 playas sampled, ten had at least one quantifiable neonicotinoid present, three of which were classified as grassland (range) playas, two were urban, and the remaining five were cropland playas (Table 3). Overall frequency of detection was 48%, regardless of playa classification. Only three playas (two cropland and one grassland) contained more than one neonicotinoid. Total neonicotinoid concentrations ranged from 0.2 to 4.2 ppb (both in grassland playas). Average total neonicotinoid concentration was 0.7 ± 0.6 ppb in grassland playas, 1.0 ± 0.4 ppb in cropland playas, and 1.1 ± 0.9 ppb in urban playas. Overall, there was no statistically significant difference in total neonicotinoid concentration among playa type.

Thiamethoxam was the neonicotinoid most frequently detected, occurring in eight of the sampled playas. Four of the samples containing thiamethoxam were from cropland playas, with two each from grassland and urban playas. Clothianidin was detected in four playas, two classified as cropland and two classified as grassland. Imidacloprid was detected solely in cropland playas; acetamiprid was only detected once, in a grassland playa, but at a relatively high concentration (3.5 ppb) compared to other neonicotinoids in our study.

The playa results from this study are fairly comparable to several published studies in similar wetland systems. A temporal study (*Main et al., 2014*) determined neonicotinoid concentrations in prairie pothole wetlands, which are similar hydrogeological features to playas, near croplands in Canada where neonicotinoid seed treatments are also heavily used. Average total neonicotinoid concentrations in the summer were below one ppb; maximum total concentrations reached over three ppb in the summer of 2012. Percent detection ranged from 16% in Fall 2012 to 91% in Spring 2013. Clothianidin was the most frequently detected, followed by thiamethoxam, differing slightly from our study.

Another wetland study, also conducted on playa lakes in the SHP in 2005, found average thiamethoxam concentrations of 3.6 ppb and acetamiprid concentrations of 2.2 ppb in crop and grassland playas (*Anderson et al., 2013*). Maximum detected concentrations of thiamethoxam were 20 ppb in cropland playas and 225 ppb in grassland playas. Maximum acetamiprid concentrations were 44 ppb in cropland playas and 27 ppb in grassland playas. While the average concentrations are somewhat comparable to those in our study, the maxima are significantly higher than those detected in this study. Frequency of detection of thiamethoxam was comparable to this study, at 31% detection in cropland playas and 25% detection in grassland playas.

Studies of neonicotinoid concentrations in other aquatic systems besides dipressional wetlands, have also been conducted. A study of groundwater wells in a potato producing region in Quebec (*Anderson, Dubetz & Palace, 2015*) found imidacloprid in 61% of samples, with a maximum detected concentration of 6.1 ppb, or approximately 3.5 times the maximum concentration found in this study.

In samples from rivers around Sydney, Australia average concentrations of acetamiprid, clothianidin, imidacloprid, and thiamethoxam (as well as thiacloprid) were all below 0.5 ppb, and as low as 0.2 ppb for thiamethoxam. Clothianidin concentrations were similar to our study at an average of 0.42 ppb, but all other average concentrations were nearly an order of magnitude lower. Frequency of detection ranged from 27% for thiamethoxam to 93% for imidacloprid (*Sánchez-Bayo & Hyne, 2014*). A 2012 study on imidacloprid concentrations in rivers and creeks in three agricultural regions of California found a maximum imidacloprid concentration of 3.29 ppb; that was the highest reported imidacloprid concentration in the US at the time. A total of 19% of samples exceeded the EPA chronic aquatic invertebrate exposure benchmark of 1.05 ppb at the time, and imidacloprid was detected in 89% of samples (*Starner & Goh, 2012*).

Both of the samples that contained imidacloprid in our study exceeded the *US EPA (2019)* chronic benchmark of 0.01 ppb. The four samples containing clothianidin exceeded the EPA chronic exposure benchmark of 0.05 ppb. The benchmark for acetamiprid is 2.1 ppb (chronic), a number that was exceeded by the lone acetamiprid detection. The aquatic invertebrate benchmark for chronic exposure to thiamethoxam is 0.74 ppb; three playa samples exceeded this metric. Overall, 28% of collected samples exceeded at least one EPA aquatic invertebrate benchmark for chronic neonicotinoid exposure. If a thiamethoxam concentration benchmark becomes available, this percentage of samples that exceed at least one benchmark will likely increase. Two of the cropland playas samples exceeded EPA benchmarks for imidacloprid acute toxicity.

When compared to results from the runoff model, the total detected neonicotinoid concentrations are all below the maximum 7.1 ppb concentration calculated. Several playas did have total concentrations approaching that level, particularly one grassland playa with a total concentration of 4.2 ppb. Because these playas were from all classifications (one urban, one grassland, and two cropland) it is difficult to draw conclusions as to the specific source of neonicotinoids. It is interesting to note, however, that for the grassland playa with the highest total concentration (4.2 ppb), a majority of the total neonicotinoid content (83%) was acetamiprid, which is not commonly used in the region as a seed treatment.

One of the key elements missing from the playa study was temporal sampling. Many of the factors that can impact neonicotinoid concentrations in playa wetlands, such as field concentrations and runoff, are extremely variable over time. Because of the dependence of playas on runoff to maintain water levels and the infrequency of major rain events in the SHP, measuring neonicotinoid concentrations through wet and dry seasons, as well as directly after major rain events, would be beneficial. In a cotton field planted with neonicotinoid-treated seeds, neonicotinoid field concentrations would naturally be higher in the late spring and early summer, corresponding to planting season. In the winter, when harvesting has been completed, and fields are not in use, neonicotinoid field concentrations would likely be lower (*Hladik et al., 2018*). Other seasonal farming practices could also have an impact on the availability of neonicotinoids in runoff.

Neonicotinoid concentrations may also vary annually; it is likely that concentrations would vary between extremely dry and extremely wet years, as amount of run-off increases or decreases, or as water volume is lost in playas due to evaporation and aquifer recharge. Years with unusually high numbers of cotton pests may also cause variation in concentrations as reapplication may be necessary. Due to these possible environmental sources of variation, an accurate depiction of neonicotinoid concentrations in playa wetlands is not possible without some temporal sampling.

## CONCLUSIONS

Neonicotinoid seed treatments appear to be an acceptable means for a more localized application of insecticides to cotton plants. Presence of imidacloprid and thiamethoxam in the leaves of plants grown from treated seeds indicates that plants successfully translocate the active ingredient through vascular tissue. Claims that seed treatments protect plants for 3–4 weeks post-planting appear to be valid based on concentrations of active ingredient present in leaf tissue and may even be an underestimation. Although seed treatments may reduce the total amount of active ingredient applied to a field and may reduce other sources of non-target exposure (such as pesticide drift), neonicotinoid residues were present in the test fields, even 150 days after planting.

The presence of neonicotinoids in soil raises concern over the potential for agricultural runoff into water bodies, particularly in the SHP where playa wetlands are common. Analysis of water samples from 21 playa wetlands showed the presence of neonicotinoids in nearly half of all samples. A total of 28% of collected samples had concentrations of at least one neonicotinoid in exceedance of the EPA Aquatic Invertebrate Benchmarks for chronic exposure. This suggests that neonicotinoids in playa wetlands have the potential to affect macroinvertebrate community densities and structure. Additional temporal monitoring of playa neonicotinoid concentrations coupled with invertebrate sampling would allow for any possible correlations between neonicotinoid concentrations and macroinvertebrate density (*Van Dijk, Van Staalduinen & Van Der Sluijs, 2013*). It may also allow for the identification of any multi-generational or delayed effects.

### Funding

This research was supported in part by Cotton Incorporated CORE funding to Dr. Apurba Barman and Texas A&M AgriLife Extension Service. The funders had no role in study design, data collection and analysis, decision to publish, or preparation of the manuscript.

### Grant Disclosures

The following grant information was disclosed by the authors:
Cotton Incorporated CORE.
Texas A&M AgriLife Extension Service.

### Competing Interests

Todd A. Anderson is an Academic Editor for PeerJ.

### Author Contributions

- Kristina L. Kohl conceived and designed the experiments, performed the experiments, analyzed the data, prepared figures and/or tables, authored or reviewed drafts of the paper.
- Lauren K. Harrell conceived and designed the experiments, performed the experiments, analyzed the data.
- Joseph F. Mudge performed the experiments, analyzed the data, prepared figures and/or tables, authored or reviewed drafts of the paper.
- Seenivasan Subbiah performed the experiments, analyzed the data, authored or reviewed drafts of the paper, approved the final draft.
- John Kasumba performed the experiments, analyzed the data, prepared figures and/or tables, authored or reviewed drafts of the paper.
- Etem Osma performed the experiments, analyzed the data.
- Apurba K. Barman conceived and designed the experiments, analyzed the data, contributed reagents/materials/analysis tools, authored or reviewed drafts of the paper, approved the final draft.
- Todd A. Anderson conceived and designed the experiments, analyzed the data, contributed reagents/materials/analysis tools, prepared figures and/or tables, authored or reviewed drafts of the paper, approved the final draft.

### Data Availability

The raw data is available as a Supplemental Information File.

### Supplemental Information

Supplemental information for this article can be found online at http://dx.doi.org/10.7717/peerj.6805#supplemental-information.

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
