# Peer review of "Tracking neonicotinoids following their use as cotton seed treatments"

_PeerJ, doi:10.7717/peerj.6805_

## Round 0.1 · original submission · Major Revisions

The quality (and quantity) of data obtained in this well designed study is appreciated. However, the manuscript cannot be accepted in its current form and requires major revisions before it can be published. In particular, more efforts should be dedicated to improving the presentation and discussion of the data (including statistical analysis). Please consider carefully the reviewers detailed suggestions for improvements.

Reviewer 1 ·

Basic reporting

The manuscript is written in clear English and easy to understand.
The literature used and background are poor and not up to date on some points. I indicate below specific points that need better explanation and citation.
The only figure presented lacks data points, while error bars are missing. The soil data could have been shown in a nice figure, and the playa data are poorly presented in a table without statistics, so comparisons with previous work are difficult.

Experimental design

The design is proper for the objectives of the study.
More water sampling on the playa would have been desirable, but I guess there were budget limitations.
The analytical methods are good, but recoveries and limits of detection are not indicated in the text - limits of detection appear in some tables though.
Unfortunately, authors missed he opportunity to analyse for clothianidin residues in both plant and soil samples; this metabolite of thiamethoxam was only analysed in the playa samples. It is a pitty, since information about this metabolite in plant material is lacking, and the sampling design here would have been very useful in clarifying this important point.
The statistical analysis is poor, not well explained in the methods and almost unreported in the tables and figures; only some p values are given in the text.

Validity of the findings

The residue data presented are valid, but the presentation is quite poor.
Comparison of the data with previous studies could be improved, in particular the residues in playa water samples. No risk evaluation is done, and the only benchmark values cited are out-of-date. Authors should consult the latest regulations by the US EPA in regard to neonicotinoids in waters.

Additional comments

line 38: rather than “nitrogen-containing”, indicate these insecticides are “chloro-nicotinic” compounds, because chlorine gives them stability and nicotine is their base.
line 39: the citation Main et al. 2014 is not very appropriate in this context; a better one would be Simon-Delso et al. 2015, Environ. Sci. Pollut. Res. 22, 5-34.
line 41: better say “…include mainly seed coatings.”
line 44-5: this statement requires a citation(s). Suggestion: Bonmatin et al. 2015, Environ. Sci. Pollut. Res. 22, 35-67. Please add also that natural waters are usually turbid and these half-lives are, therefore, longer - check the IUPAC database on pesticides, http://sitem.herts.ac.uk/aeru/iupac/atoz.htm
line 51: that regulation by the European Community was superseed last year; see Official Journal of the European Union L 132, whereby the same three compounds were banned from application outdoors - they can only be applied in glasshouses.
line 57: in this context, please cite Sanchez-Bayo et al. 2016, Front. Environ. Sci. 4, 71
lines 57-9: no, Hallmann et al. 2014 did not report on migratory waterfowl, but on small insectivorous birds! Waterfowl usually do not feed on insects but on algae, grass and other plant material.
line 61: yes, neonicotinoids are more selective not only for pests but also for parasitoids and natural enemies, unfortunately. Please indicate this drawback, because it is one of the criticisms made of this class of compounds, and the reason why they are not welcomed in IPM programs.
lines 68-70: this comment should be backed up by citing the work by Balfour et al. 2016, Agric. Ecosyst. Environ. 215, 85-88.
line 81: sorry, but authors assume that only chemicals can offer protection against early pests, whereas the reality is different. Ladybirds and other natural enemies are the best protectors of the crop at these early stages, so if we use neonicotinoids as seed treatments, they are eliminated completely. Please indicate here that there are other ways of containing the early outbreak of aphids and other suckers. You can refer to Furlan et al. 2018, doi:10.1007/s11356-017-1052-5
line 105: how far apart were the treated rows?
line 106: are pivot sprinklers the most common irrigation system in Texas? This may not be representative of the common practices for this crop, although it is OK for the purposes of the study. Other countries prefer flood irrigation, which will move the residues in the soil among all plants, treated or untreated.
line 108: insert a subheading “Sampling”
line 118: indicate how the soil was collected, i.e. auger, hand-held core, scoop…I don’t understand what “Eight cm” of soil means in this context; shouldn’t be “Eight cm deep core” or equivalent? What was the approximate mass of the samples?
line 132: were the water samples collected in amber or dark containers/bottles? If not, the residues analysed would be compromised by photolytic degradation in the containers.
line 234: insert a subheading “Sample analysis”
line 136: this procedure is for determining the dry soil weight, so indicate its purpose here.
line 133: the expected analytes would be the two applied (imidacloprid and thiamethoxam) plus the metabolite clothianidin. I’m surprised to see that authors analysed the metabolite in water samples but not in the others. This is a missed opportunity, since the same analytical effort is necessary for analysing two or three compounds in a given sample. In particular, it would have been interesting and novel to report the levels of clothianidin in the leaves from thiametoxam-treated plants…
line 174: apart from the analytical methods, a summary of extraction recoveries for each matrix (leaves, soil and water) should be presented. Also, limits of detection and quantification for both HPLC and LC/MS methods should be indicated either in the text on in the respective tables.
lines 176-80: this section is poor. What comparisons were made? For instance, how would you compare the levels of the residues among the three playas? Student t-tests are not applicable in this case. Also, did you test the assummptions of normality of the data distributions? Residue levels of chemicals in environmental samples usually follow logarithmic distributions, so transformations of the data are necessary to achieve normality and use statistical tests such as the t-test. Furthermore, the soil data allows for estimation of a rate of disappearance from the field, as the late samplings were done about 150 days after planting of the seeds. This information is important, and is typically expressed as half-lives.
line 187: what about clothianidin?
lines 192-3: to make sense of these concentrations you must refer to the days after planting; e.g. 31-36 days for the 18 ppb values and 21 days for the 10,600 ppb.
lines 194-220: these paragraphs should be summarised and refer to the days after planting in order to understand the differences in residue levels.
lines 224-5: along the same lines as above, indicate in the text the difference between high and low values. Calculation of a dissipation rate (half-live) with time is feasible: you only need to combine all the data points from the various locations, as shown in Figure 1, and plot them against the number of days after planting.
lines 231: alternatively, you can also calculate the rate of residue change between the two sampling dates at the same locations.
line 255: please, cite here Balfour et al, 2015.
line 261: “previously reported” requires a reference.
lines 270-93: as for the residues in leaves, please summarise the findings and refer to the days after planting to understand the dissipation of the soil residues among the various locations.
lines 314-5: this is an important finding, and since 80% of the coating remains in the soil (Goulson 2013), it should be stressed here that the seed-treatment results in a long-term contamination of the field, which for compounds like imidacloprid appear to accumulate year after year (see Jones et al. 2014, Pest Manag. Sci. 70, 1780-1784); thiamethoxam may not accumulate because it is converted to clothianidin, but this metabolite is very persistent in soil - see in this regard Schaafsma et al. 2016, Environ. Toxicol. Chem. 35, 295-302. and Limay-Rios et al. 2016, Environ. Toxicol. Chem. 35, 303-310.
line 317: indeed, that appeared to be the case in the experimental work reported by Sanchez-Bayo et al. 2007, J. Environ. Sci. Health B 42, 279-286.
line 317-9: degration rates may vary among the two conditions you mention, but in the case of very soluble compounds like neonicotinoids, it is not degradation as such (i.e. chemical decay and biological metabolism) that is crucial but the actual dissipation rates due to losses by leaching and runoff. These two factors prevent accumulation of residues in the field to a certain point, because half of the amounts applied as coatings will dissolve in water after a rain or irrigation event and move into the nearby drains or percolate (ie. leaching) through the soil profile and contaminate the ground water - see Gupta et al. 2008, Bull. Environ. Contam. Toxicol. 80, 431-437.
line 322: please cite here Jones et al. 2014
line 323: it is not only a “possibility” but rather a fact; there is plenty of evidence of this, and for the USA please refer to Hladik and Kolpin 2014, Environ. Pollut. 193, 189-196.
line 324: as said above, not just run-off, but mainly dissipation by leaching - see also Wettstein et al. 2016, J. Agric. Food Chem. 64, 6407-6415.
lines 335-41: this paragraph should be moved into the Methods section.
line 344: better indicate the actual water solubility of the two compounds used in this trial, which is well known and reported.
line 249: insert a subheading “Playa samples”
line 351: indicate the actual percentage of positive samples (47%) instead of an approximate number.
line 366: please explain here the reason for adding clothianidin to the list of analytes of the playa samples, whereas it wasn’t included in the soil or leaf samples. After all, the contamination of the waters comes from the soil and perhaps leaves.
lines 371-81: these statistics should be included in Table 3 as well - see comment below.
line 418: the current EPA acute bechmark values for imidacloprid is 0.385 ppb; the chronic benchmark is 0.01 ppb.
line 422: there is an acute benchmark for thiamethoxam in the USA, established as 17.5 ppb by the EPA - please, update your reference sources.
lines 425: obviously, some of the samples exceed the current EPA benchmark values.
lines 435: temporal sampling was obtained in this study for the soil residues.
lines 443-5: please, cite here Hladik et al, 2018, Environ. Pollut. 235, 1022-1029.
line 445: in fact, tilling may increase the leaching potential of these soluble compounds, whereas the more sustainable practice of non-tilling reduces their leaching potential.
line 462: see comment above in regard to aphids: there is hardly any need to apply neonicotinoids for controlling aphids when natural enemies are used under IPM programs; let alone for re-application.
line 467: this is a rather misleading statement; as mentioned above, IPM practices may render neonicotinoid application unnecessary and in fact damaging to the IPM itself.
line 475: add to the sentence “…and move into waterways nearby and groundwater.”
line 480: please revise your conclusions after taking into account the current EPA benchmarks.
line 486: you could cite here the correlaton study by vanDijk et al. 2013, PLoS One 8, e62374.
Table 1: suggest adding a column to indicate the number of days after planting
Table 2: these data would be better shown in a figure, plotting the average residues (plus error bars) against the days after planting. In that way we can visualize the trend with respect to the application day.
Table 3: the raw data in this table are not helpful; a summary of the detections, average and highest levels, and statistical differences among playa types would be more meaningful. You may need to swap the rows and columns to present the data in this way.
Figure 1 presents the same data as in Table 1 except that a couple of data points apperar to be missing for each compound - there are 8 different data point in Table 1, but only 6 here. Error bars are also missing.

·

Basic reporting

The article is well-written. It provides a clear background and justification for the study and pertinent literature for context. The manuscript presents good data obtained from diverse media, insecticides and scenarios with high relevance in meeting the aim of the work.

Experimental design

The article fits well within the aims and scope of PeerJ.
The aim of the manuscript is clearly defined and relevant. Even more relevant and elaborate is the research design, thereby ensuring the aims are met. The relevance of this article to a clearer understanding of the fate of neonicotinoids is not in doubt. The experimental design appears well-thought through and represent good quality methodology. The use of LC-MS was revelant for detection of very low quanitities of insecticides as reported in the work.
The sampling procedure is clearly explained for the various types of samples: leaves, soil and water.
My comment is on the sampling proceedure for soil:
The authors explanation of sampling 15 cm from the plant makes a lot of sense. However, if the aim is to track neonicotinoids following their application as seed treatment, then leaving out a potentially large concentration of the applied insecticide possibly present in the seed coating or remains, may not give a good estimate of exposure in the environment. Possibly, a suggestion to the effect that, potentially large amounts of insecticide may be present around location of the seed will be help enrich the study.
Sur & Stock (Bulletin of Insectology 56 (1): 35-40, 2003) has shown that <5% of imidacloprid used in seed treatment in cotton is taken up by plants.
Such an estimation based on knowledge of concentration applied neonic and concentration found in cotton leaves following germination will be a good idea.

Validity of the findings

There is a great deal of data presented which is good and provides good validation for the study. However, this advantage of a large pool of data makes reading of the results (particularly without ready access to tables) rather difficult and boring. while I belive this will improve when the final article is ready, generalised discussion highlighting the major differences or similarities in results rather than the mention of almost every single value for the various pesticides, media sampling dates and sampling sites may be ideal.

The authors have provided a good discussion of the results obtained and provided sufficient reasoning or plausible explanation for findings. The data is well analysed statistically and the inclusion of models to further predict fate in the environment in commendable.
In table 3, detected limit is indicated as 0.05 ng/mL. However quantification limit is not inidicated. It would be important to indicated the LoQ as values presented are based on quantification instead of detection. This is expectially because values of 0.1 and 0.2 ng/mL (very close to LoD) have been reported as quantified (for clothianidin).

Additional comments

As indicated by the authors, the scope of work was broadened as work progressed further enriching the manuscript. However, I believe a clearer background in this respect would be ideal here.
For instance, while the study was focused on imidacloprid and thiamethoxam, other neonicotinoids were analysed for in water samples. This background was lacking and hence the first appearance of clothianidin or acetamiprid outside of literature appears out of place.
Kindly edit the following:
line 24 "as a result in" may read "as a result of" or "may result in"
line 55. consider replacing the word charismatic
line 57 & 58 may read "A possible connection between declines in migratory waterfowl, reduced aquatic invertebrate populations along migratory pathways and neonicotinoid use..."
line 64: but the issue may read. although this assertion...
line 77-79: particularly imidacloprid and thiamethoxam may come right after neonicotinoids in line 77
line 166: conducted instead of conducting
line 171: used
Consider moving reference in line 388 tp 385
line 467: consider "for" instead of "of"
line 472: consider underestimation

---

## Round 0.2 · accepted · Accept

We appreciate your careful consideration of the reviewers´comments, which helped to improve the manuscript.

#